# Venom in Furs: Facial Masks as Aposematic Signals in a Venomous Mammal

**DOI:** 10.3390/toxins11020093

**Published:** 2019-02-05

**Authors:** K. Anne-Isola Nekaris, Ariana Weldon, Muhammad Ali Imron, Keely Q. Maynard, Vincent Nijman, Stephanie A. Poindexter, Thais Queiroz Morcatty

**Affiliations:** 1Nocturnal Primate Research Group, Faculty of Humanities and Social Sciences, Oxford Brookes University, Oxford OX3 0BP, UK; a.weldon@brookes.ac.uk (A.W.); keely.maynard@mail.utoronto.ca (K.Q.M.); vnijman@brookes.ac.uk (V.N.); spoindex@bu.edu (S.A.P.); thais.queiroz.morcatty-2018@brookes.ac.uk (T.Q.M.); 2Faculty of Forestry, Universitas Gadjah Madah, Yogyakarta 55281, Indonesia; maimron@ugm.ac.id; 3Department of Anthropology, Boston University, Boston, MA 02215, USA

**Keywords:** colouration, concealment, crypsis, intraspecific competition, *Nycticebus*, predation, visual system

## Abstract

The function of colouration in animals includes concealment, communication and signaling, such as the use of aposematism as a warning signal. Aposematism is unusual in mammals, and exceptions help us to understand its ecology and evolution. The Javan slow loris is a highly territorial venomous mammal that has a distinctive facial mask and monochromatic vision. To help understand if they use aposematism to advertise their venom to conspecifics or predators with different visual systems, we studied a population in Java, Indonesia. Using ImageJ, we selected colours from the facial masks of 58 individuals, converted RBG colours into monochromatic, dichromatic and trichromatic modes, and created a contrast index. During 290 captures, we recorded venom secretion and aggressiveness. Using Non-metric Multidimensional Scaling and generalised additive models for location, scale and shape, we found that young slow lorises differ significantly from adults, being both more contrasting and more aggressive, with aggressive animals showing fewer wounds. We suggest aposematic facial masks serve multiple purposes in slow lorises based on age. Change in colouration through development may play a role in intraspecific competition, and advertise toxicity or aggressiveness to competitors and/or predators in juveniles. Aposematic signals combined with intraspecific competition may provide clues to new venomous taxa among mammals.

## 1. Introduction

The evolutionary function of colouration in vertebrates includes physiological aspects, concealment, courtship, species recognition, and intra- and interspecific communication and signalling [1,2]. The latter includes the use of aposematism (bright or conspicuous colouration) as a warning signal to deter predators or to emphasise an animals’ unpalatability, which includes toxicity [3,4]. Multimodal warning systems, including colouration and defensive weapons, are common in aposematic animals, and might help to explain the evolution of this paradoxical trait, whereby predators or competitors need to learn to avoid a dangerous encounter [5]. Aposematic colouration also may be used as a form of masquerade through mimicry of other toxic species [6], and the ultimate mimicry of one or more defensive properties, including aggressive behaviour, warning calls, unpalatability or toxicity [7]. For species that produce strong olfactory signals or toxins, visual conspicuousness may have evolved because of their profitability in warding off competitors or predators, as in seen a number of insect species [8]. In the case of mammals, stripes or patches may point towards areas of the body that deliver these toxic defences [3].

Among mammals, aposematism has been seen as one mechanism to explain contrasting black and white colouration, with other mechanisms including physiological functions and interspecific communication [9]. Reviewing more than 5000 terrestrial mammals, Caro [3] examined the function of conspicuous black and white markings on facial masks, the chest, neck, dorsum or tail. In some primates and ungulates, eye rims may have a thermoregulation effect or an anti-glare function. In other primates and ungulates, including lemurs (Lemuridae, Cheirogaleidae) and dik diks (*Madoqua* spp.), bright markings were associated with scent glands and may be used to reinforce how these areas are used in intraspecific communication. In giant pandas (*Ailuropoda melanoleuca*), black and white markings may serve multiple functions, including background matching and intraspecific communication, including emphasising ferocity [10]. Mid-guild carnivores, including badgers (*Meles* spp.), skunks (*Mephites* spp.), and zorillas (*Ictonyx* spp.), which may be subject to predation by larger carnivores, use aposematism to warn predators of their aggressive behaviour, odour or unpalatability [1,11].

Different forms of markings may have a range of not mutually exclusive functions, depending on how the signal is perceived and processed, including by receivers with different colour vision [6]. The RGB [red-green-blue] colour space is a standard system used to quantify colours within the human visual spectrum [12]. The blue-yellow channel is found more universally among mammals, while the red-green channel is more limited [13]. Thus, the function of colour may differ according to a perceiver’s visual system. For black and white patterns, the function of contrast alone may be important. Visual contrast is a key aspect of visual perception [14], due to variations in an animal’s ability to interpret red, green, and blue within its environment. Contrast is, thus, a universally accessible visual cue. Despite the wide array of visual cues (e.g., red-green levels, blue-yellow levels, luminosity, contrast), only some are processed in detail and contrast is one of the easiest and most salient visual cues to interpret [15]. It is thus important to understand how features including RGB colour, contrast, hue, modality and patterning might affect the behaviour of viewers including predators, group mates and competitors [16]. For example, the same patterns that serve as aposematic signals against predators may also serve to attract mates [17], communicate with rivals [18], or match backgrounds [10,19]. Many taxa are largely cryptic but can reveal their aposematic signals when threatened or startled [19]. Juveniles, in territorial species, can also have vibrant or more conspicuous patterns to signal their age to adult individuals and thereby reduce aggression [20]. This more extreme signal in adolescence can also possibly serve as an anti-predator defense as well by increasing the novelty of the prey to the predator and decreasing the chance of predation [16].

Found throughout Southeast Asia, slow lorises (*Nycticebus* spp.) are slow moving, medium-sized nocturnal primates with highly contrasting facial masks [21]. A multitude of criteria point towards the use of the facial mask as aposematic signal [9]. Slow lorises are one of only seven taxa of venomous mammals, and the only venomous mammal to possess a striking facial mask. Their venom is produced through combining oil from a brachial gland with saliva from their mouth when the animal raises its arms over its head [22]. When threatened, the slow loris hangs bipedally with the arms folded above the head, meaning that the nuchal area is most likely presented to a combatant [23]. Their venom was initially proposed ultimately to have an anti-predator function [24]. Predation on lorises, however, has rarely been observed, with a handful of predation events limited to a few taxa with different visual systems: hawk eagles (*Nisaetus* spp.), pythons (*Python* spp.), monitor lizards (*Varanus* spp.), and orangutans (*Pongo* spp.) [25]. In this study, we hypothesised that mask colour and the use of venom plays a more important proximate role in intraspecific competition, by signalling toxicity or the potential for a higher level of aggression.

Javan slow loris (*N. javanicus*) adults show no sexual dimorphism and form uni-male, uni-female social groups in highly defended territories, yet exhibit a promiscuous non-seasonal mating system [21]. The facial masks of young and old Javan slow lorises (*N. javanicus*) are strikingly different, based on a human trichromatic visual system; they were once described as two different species [26]. Slow lorises can kill each other with their venom [27], and thus a signal that advertises this weapon could potentially facilitate conspecific avoidance. Although all age classes of slow lorises fight, conflicts with subsequent (potentially fatal) wounds are especially frequent during dispersal when younger animals attempt to settle into a new territory [22,27]. Younger animals are more pugnacious and one of the few documented cases of near fatality in humans came from the bite of a juvenile slow loris [28]. Greater levels of colouration or contrast may quantitatively indicate the level of toxicity [29]. In younger animals that may either be more toxic or unable to meter their venom [30,31], we hypothesised that a more contrasting facial mask may advertise their toxicity.

Here, we aim to understand whether colouration patterns affect how potential predators and competitors perceive slow lorises, and whether any detected differences have a functional significance, particularly as an anti-predator advertisement or for use in intraspecific communication. We measured colour characteristics of the facial masks of known individuals from a single population of Javan slow loris. We tested if age classes could be differentiated by colour from the perspective of animals with different colour vision (monochromatic, dichromatic, trichromatic). To examine whether colour or age were related to toxicity, we recorded aggressiveness and gland secretions during capture as a proxy for toxicity. We predicted that if the facial mask has an anti-predator function, more toxic individuals would be distinguished in dichromatic or trichromatic colour schemes. We predicted that if slow loris facial masks signal intraspecific recognition or competition, age classes would be distinguished by differences in colour or contrast in the monochromatic scheme. Finally, we predicted that if the facial mask advertises higher degree of toxicity, animals that had more gland secretions will show the most contrasting facial masks. We discuss these results in relation to the function and evolution of mammalian coat colour and the evolution of venom use in intraspecific competition.

## 2. Results

### 2.1. Colour

Considering the colour values in Munsell system in a monochromatic scheme, we found no separation among age classes in the Non-metric Multidimensional Scaling (NMDS) (Figure 1) and no relationship between the colour values and the individual ages in days (E-value (E) = −0.008, Standard Error (SE) = 0.0006, *t*-value = −1.39, coefficient of determination (r^2^) = 0.08, *p* = 0.17). In both dichromatic and trichromatic schemes, we found no separation among age classes in the NMDS (Figure 1) (*p* < 0.05) and no relationship between the colours and the individual ages in days (E = −0.236, SE = 0.051, *t*-value = −4.62, r^2^ = 0.01, *p* = 0.63 and E = −0.0039, SE = 0.0005, *t*-value = −0.522, r^2^ = 0.016, *p* = 0.60, respectively).

Conversely, we found a difference in the Contrast Index (which includes contrast ratio, relative luminance difference, brightness difference and colour difference) among age classes. Infants differed from all other classes, and adults and juveniles significantly differed from each other, while subadults were intermediate between adults and juveniles, overlapping totally with both groups (*p* < 0.01; r^2^ = 147) (Figure 2). We found a significant relationship between the Contrast Index ordination (Axis-2) and the individual ages in days (Figure 2) (E = −1.55, SE = 0.0001, *t*-value = −5.06, r^2^ = 0. 54, *p* < 0.001).

### 2.2. Aggressiveness

We found several important significant relationships regarding level of aggressiveness (Figure 3 and Figure 4, Table 1). Although both sexes were aggressive, males were significantly more aggressive than females (Table 1). Across both sexes, more aggressive animals displayed fewer wounds (Figure 4e). The level of aggressiveness decreases with an increase in the individual body size; juveniles to subadults were the most aggressive. We found no relationship between brachial gland secretion and aggressiveness, but we did find that animals that had been caught more often became less aggressive over time. Individuals with greater contrast index values were more aggressive or feisty than individuals presenting lower contrast index values (E = −0.010, SE = 0.0045, *t*-value = −2.29, *p* = 0.03) (Figure 3).

## 3. Discussion

Here we present the first testing of the adaptive significance of colouration patterns in a venomous mammal. We found evidence that the facial mask of slow lorises serves as an aposematic signal. Although colours themselves did not distinguish age classes, under any of the three visual systems tested, contrast did. In particular, younger animals had more contrasting facial markings than adults. This trait is potentially perceivable by animals with any of the three visual systems tested, including other slow lorises. We also found that the most aggressive animals were males in the juvenile to subadult body range. These more aggressive animals showed higher levels of contrast and were less likely to have fresh wounds. However, the amount of brachial gland fluid secreted did not relate with any of the variables. Several explanations may account for why contrast distinguishes age classes in slow lorises, and we focus on three of these below: a cryptic anti-predator function, intraspecific competition, younger animals advertising a higher degree of toxicity.

### 3.1. Cryptic Anti-Predator Function

Our prediction that potential dichromatic and trichromatic predators would be able to distinguish slow loris individuals by colour was not supported, because colours themselves did not distinguish age classes under any of the three visual systems tested. Yet, such potential predators still could perceive contrast. Slow lorises are highly cryptic, communicating in the ultrasonic range and moving slowly without leaping or disrupting vegetation [32]. If this cryptic strategy should fail and the loris is spotted by a predator, possession of a powerful aposematic signal could be an advantage. In mammals, eye spots and eye patches may help an animal to appear larger to a potential predator and, if black and white, they are more easily abstracted [33]; slow loris features may serve this function. The dark markings of the facial mask disrupted by white markings may be a form of weak conspicuousness [34]. Potential predators of slow lorises, including lizards, snakes and birds, all may vary in their ability to ignore unpalatibility or avoid toxicity. Thus, in their evolutionary past, the relative importance of different predators might have been related to the evolution of aposematic signals in slow lorises [35,36]. Further studies on predation risks are presently needed to test this hypothesis.

### 3.2. Intraspecific Competition

Our prediction that age classes could be distinguished by contrast was supported. In addition, individuals with a more contrasted mask were more aggressive. It is not uncommon for aposematic signals to serve multiple functions [37]. Thus a facial mask that advertises toxicity to potential predators can serve a similar function towards conspecific competitors. In particular, strongly contrasting black and white signals at night are more likely to be used as signals towards conspecifics [3]. We found that the contrast of facial masks was highly distinguished between age classes, especially between lorises of dispersing age and adults. Javan slow lorises largely live in uni-male, uni-female social groups with 1–3 offspring in heavily defended territories [21]. From our long-term field work, we know that pairs can maintain these territories for up to eight years, meaning that the establishment of a permanent territory is a vital component of slow loris development [32]. Distinct facial features may communicate a warning signal, indicating aggressiveness of young animals looking to settle in a permanent home range, such as also seen in sciurids [2] and blue-ringed octopus (Hapalochlaena lunulata) [38].

Other taxa use visual signals for social cohesion or as warnings for intraspecific competition. For slow lorises, advertising their toxicity with bold black and white colouration parallels that of many small carnivores, including stink badgers (*Mydaus* spp.), striped polecats (*Ictonyx striatus*), marbled polecats (*Vormela peregusna*) and grisons (*Galictus* spp.) [11]. Many small mid-guild carnivores, including those that are slow moving and stocky like slow lorises, have a facial mask and discharge toxic spray, suggesting that their facial masks warn and deter predators and competitors in a similar niche [11]. Indeed, several features of slow lorises meet the criteria of Stankowich et al. [4] regarding advertising other animals to stay away, aiding in avoiding potentially dangerous altercations [11]. First, the location of the patterns is near the venom glands of the Javan slow loris (with stripes pointing to the mouth). The slow loris’ stripes converge on the top of the head and at the neck, which is associated with toughened skin, allowing the slow loris to be wounded, but still escape the competitor’s grip, such as also seen in badgers (*Meles* spp.), otters (*Lutra* spp.), wolverines (*Gulo gulo*) and zorillas [11]. Indeed, slow lorises most often bite the head, and of 76 wounds in our sample, 32 were in this region (Figure 4f). These regions were also more likely to recover leaving only a scar, whereas more vulnerable bitten areas like ears and digits were lost in bitten victims. As a signal to competitors, more aggressive animals are those that showed the least number of wounds, and thus the facial mask may serve to be an honest signal showing the quality of venom [8]. Contrary to our prediction that more toxic individuals would have a higher level of brachial gland secretion, this relationship was not a statistically significant relationship. However, it is important to note that slow loris saliva also contains venom and that the saliva and brachial fluid may serve separate toxic functions, with the former associated with pain and necrosis and the latter associated with an anti-parasitic function [39]. To examine if the facial mask is a true signal for toxicity, better quantifiable measures of slow loris venom are needed.

### 3.3. Colour Advertising Young Animals

Our prediction that more aggressive animals would show more contrasting masks was supported. More aggressive individuals were also of younger age classes. In many animals, colouration of young animals differs markedly from that of adults [40,41,42]. The extreme black and white contrast of the Javan slow loris facial mask was also the most pronounced in juvenile animals. Although wounds were distributed across age classes, juvenile animals were the most aggressive, fiercely attacking measurers when handled, and often growling or even shrieking. Such behaviour is also exhibited by juvenile Norwegian lemmings *Lemmus lemmus*, that lack the attack power of an adult, but especially use loud calls as a form of mimicry [43]. Taitt [44] suggested that Norwegian lemmings use a strategy of crypsis when resident and aposematism when migrating, being both noisier and more aggressive during migration. This strategy mirrors slow lorises that tend to avoid conflict when settled in a permanent territory (K.A.I. Nekaris, unpublished data).

This extreme advertisement of black and white contrast in animals of dispersal age could be linked to a higher degree of aggression in young slow lorises and act as a general warning of toxicity, such as seen in other animals including rattle snakes (*Crotalus* spp.) and racers (*Coluber constrictor*) [30,45]. The contrast can create an illusion making young slow lorises appear larger [46]. As animals age and grow less vulnerable, such patterns may still be present, but become less distinct [40]. Subadults may thus retain a still relatively high degree of contrast and aggressiveness during the dangerous period of dispersal.

## 4. Conclusions

This work represents an important case study of the function of pelage colour and contrast in a venomous animal, with specific reference to the facial mask. Ecological studies of the function of colouration are crucial for understanding the evolution of aposematic strategies [16]. Slow lorises are the only venomous mammal that displays aposematic signals as well as the only known venomous primate [22]. Many other nocturnal primates have a facial mask, but lack the extreme contrast seen in the slow loris facial masks [47]. Here we found that young lorises had more contrasted facial masks and were more aggressive than adults. This supports the hypothesis that colouration patterns act as a form of communication among conspecifics. These contrasts could potentially also be perceived by predators, although further studies on predation risks are now needed. As a potential signal to advertise to predators, the facial mask seems to be effective at communicating a warning signal, as across several long-term field studies, few instances of slow loris predation have been reported [48]. The use of venom for intraspecific competition is rare among animals [49] but is characteristic of two and possibly three of the seven known venomous taxa (slow lorises, platypus *Ornithorhynchus anatinus*, and possibly solenodons *Solenodon* spp.). Mammals with aposematic face masks and markings may prove a key group to discover new venomous taxa.

## 5. Materials and Methods

From June 2011 to June 2018, we collected data on facial masks and aggressiveness of wild Javan slow lorises in Cipaganti, Garut District, West Java, Indonesia (S 7°6′6–7°7′07, E 107°46′0–107°46′5). The recorded animals occurred at elevations ranging from 950–2300 m above sea level. The habitat varies from open agroforest characterised by fairy duster (*Calliandra calothrysus*), string bamboo (*Gigantochloa apus*), green wattle (*Acacia decurrens*), avocado (*Persea americana*), to closed canopy secondary forest, characterised by johpan trees (*Lepisanthes rubignosa*) and Chinese pop tree (*Macropanax dispermum*). The climate is aseasonal, with temperatures in the day averaging 22.6 °C (range 12.4–28.0 °C), and 18.9 °C (range 12.6–26.7 °C) at night [50]. A range of potential predators is available at the site, ranging from carnivores to raptors to snakes, yet we have not observed a single incidence of predation since the study began.

From April 2012–December 2017, we captured slow lorises for radio collaring and health checks 290 times, with no need for anaesthesia [51]. We weighed animals with a Pesola spring scale. During each capture, we recorded the animal’s behaviour and venom gland secretion both on an ordinal scale. We coded behaviour as calm (the animal rested on the measuring surface in a relaxed posture with no distress calls), feisty (the animal was restless and sometimes exhibited growling or squirming at the beginning of measuring, but settled to calm), and aggressive (the animal growled, hissed or screamed; attempted to bite; and did not settle during the measuring process and required firm handling). We coded brachial gland secretion as dry (no fresh oil secreted from the brachial gland, or it was crystallised) or wet (one or both of the brachial glands exuded fluid) and recorded the presence of fresh wounds during each capture. To photograph the lorises’ faces, we held animals securely and took photographs from the same angle using a Canon 7D, Canon 5D or Nikon D700, with a 2.8 lens. We set ISO at 1600, with the aperture at 2.8. When necessary, we enhanced colour by comparing photos to a standard Munsell colour chart taken with the photo. We defined the characteristics of facial masks and crown following Munds et al. [52]. We used photographs of 39 slow loris individuals with known life history and known birth date. We defined age classes as follows: infants—≤153 days old (*n* = 3, age when they are dependent on mother and relatively immobile); juveniles—154–365 days old (*n* = 20, age when they are exploring their natal range); subadults—366–730 days old (*n* = 13, age when they make larger forays from their natal range or disperse); adults—≥731 days old (*n* = 22, actively dispersing or settled) [51]. While we had 52 individuals with photographs taken and 39 individuals (summing 150 captures events) with the behaviour and all additional traits recorded, we have just 31 individuals with all variables completed. All research was approved by the Animal Care Subcommittee of Oxford Brookes University Research (12OxbansceA2719: initially issued September 2011 and renewed annually) and followed the International Primatological Society (IPS) Guidelines for the Use of Nonhuman Primates in.

### Data Analysis

To account for the ability of different species to perceive different levels of colour receptors, for each trichromatic image taken with a camera, we made dichromatic and monochromatic copies using the Colbindor Coblis colour blindness simulator (http://www.color-blindness.com/coblis-color-blindness-simulator/). We then processed the three colour versions per individual using ImageJ software (1.52e, Public Domain, Bethesda, MD, USA). We selected ImageJ as it has the capability to copy and paste selections to all three images, ensuring that the compared sections were identical across each individual’s image. ImageJ has been similarly used to quantify animal-background contrast [53] and in colour change and camouflage of ghost crabs (*Ocypode ceratophthalmus*) [54]. Using ImageJ’s multi-point tool, we took a colour selection from the following five facial mask features: crown, interocular stripe; right and left of patch top, eye contour, and patch bottom (Figure 5). We then used the colour picker tool to select the centre of each facial mask point, and recorded RGB for each location on each colour view [53]. To determine contrast, we measured interocular stripes at their most central point, where the stripe is widest, and patch tops were measured between the top of the eye rim and the narrowest point of the patch top. Using the *Analyse > set measurements* command, we recorded mean RGB greyscale value, modal grey value, minimum and maximum grey values and median grey values for each section. We drew sections using the area selection tool and compiled into lanes for analysis, with three lanes per image: interocular stripe, patch top left and patch top right.

We consider as features of the facial mask two categories of colour measurements. First we measured the RGB colours from eight independent points on the facial mask. From this, we calculated colour intensity, by averaging the RBG colours for each point. Next, we calculated a Contrast Index in the monochromatic scheme, which is the result of a NMDS analysis, an ordination putting together all variables related to contrast (contrast ratio, relative luminance difference, brightness difference and colour difference). To use the RGB colours as a continuous gradient we converted them into XYZ-coordinates based on the Munsell colour system (e.g., Ref. [55]).

To assess whether there is cluster formation of face colour features (in monochromatic, dichromatic and trichromatic schemes) according to age classes, we used NMDS analysis based on the Bray-Curtis dissimilarity index [55] and an Analysis of Similarity (ANOSIM) to obtain the significance of the difference between these groups (*n* = 52). Afterwards, we used a non-linear model as a post-hoc test to evaluate the relationship between the two first Axis ordination from the NMDS and the individual ages in days since birth.

We used generalised additive models for location, scale and shape (GAMLSS) [56] to test the relationship between the contrast index and the level of aggressiveness (*n* = 31). We also used the GAMLSS to assess additional variables that influence the level of aggressiveness of sampled individuals, namely sex, body weight, age, contrast index, presence of wound and brachial gland secretion (*n* = 39). We included the individual as a random factor and included the number of recaptures as a variable in the model. Since, we found collinearity between body weight, age and contrast index, we did not include these variables in the same model. Instead, we used age in days as a proxy for both and presented the result for this variable, which provided a more robust model due to the high number of samples. We selected final models and the appropriate family of distribution based on the Akaike Information Criteria (AIC), the lowest AIC considered as the best best-fitted model. ΔAIC was given based on the difference between the model with the lowest AIC and the second best-ranked model.

We used the R-package munsellinterpol to convert the RBG colours in XYZ-coordinates based on the Munsell colour system, the R-package vegan to do the build the NMDS and ANOSIM, the R-package gamlss to test the generalised linear and additive models and R-package GGally to test the collinearity among variables. We assumed significance at *p* < 0.05 and stress < 0.09 as an excellent and stress < 0.20 as a good representation for reduced dimensions in NMDS. For all analyses, we used R programming (v. 3.5.1, GNU General Public License, New Zealand).

## Figures and Tables

**Figure 1 toxins-11-00093-f001:**
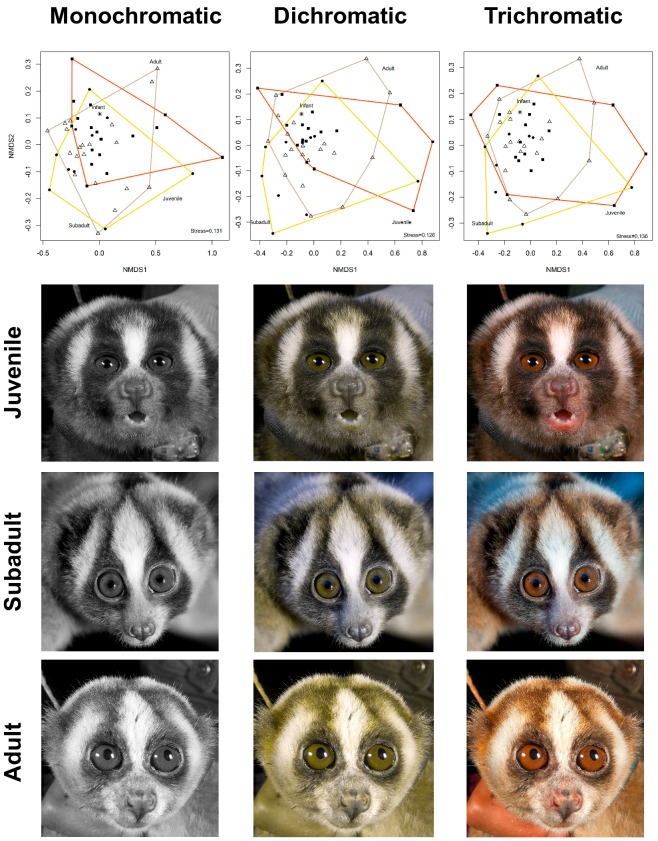
Relationship between the Munsell scale face colours in monochromatic, dichromatic and trichromatic scheme with individual age classes (∆ = Adult, ● = Sub adult, ■ = Juvenile, * = Infant) in a Non-Metric Multidimensional Scaling (NMDS).

**Figure 2 toxins-11-00093-f002:**
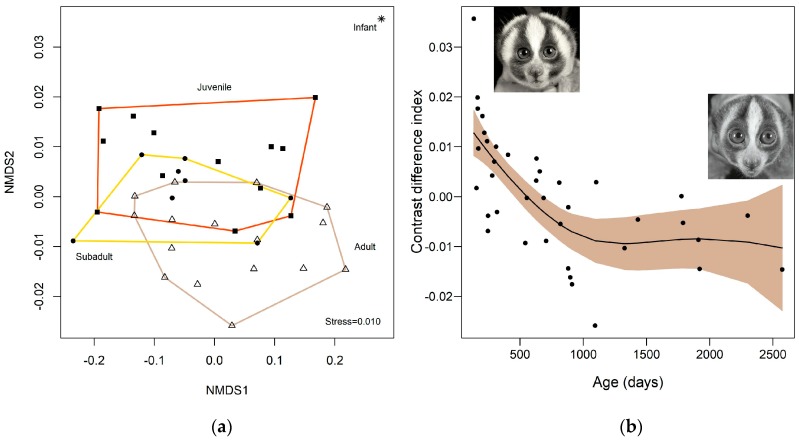
(**a**) Relationship between the face contrast features (contrast ratio, relative luminance difference, brightness difference and colour difference) in monochromatic scheme with age classes in a Non-Metric Multidimensional Scaling (NMDS) and (**b**) relationship between the Contrast Index ordination (Axis-2) and the individual ages (in days). The coloured area represents the 95% confidence level interval and Y-axis are expressed as partial residuals related to the mean (µ = 0).

**Figure 3 toxins-11-00093-f003:**
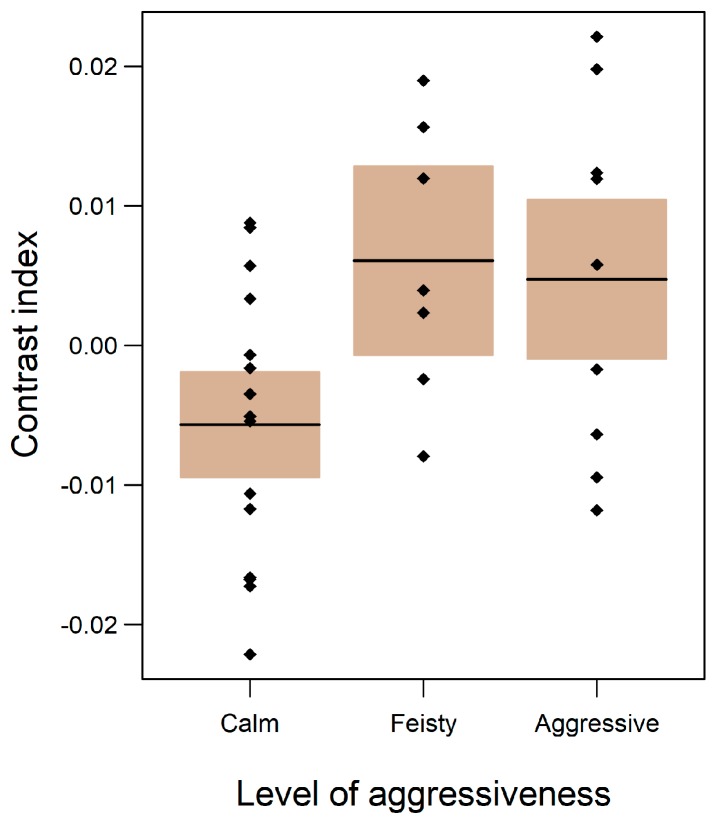
Relationship between contrast index and level of aggressiveness (*n* = 31). The coloured area represents the 95% confidence interval and the Y-axis is expressed as partial residuals related to the mean (µ = 0).

**Figure 4 toxins-11-00093-f004:**
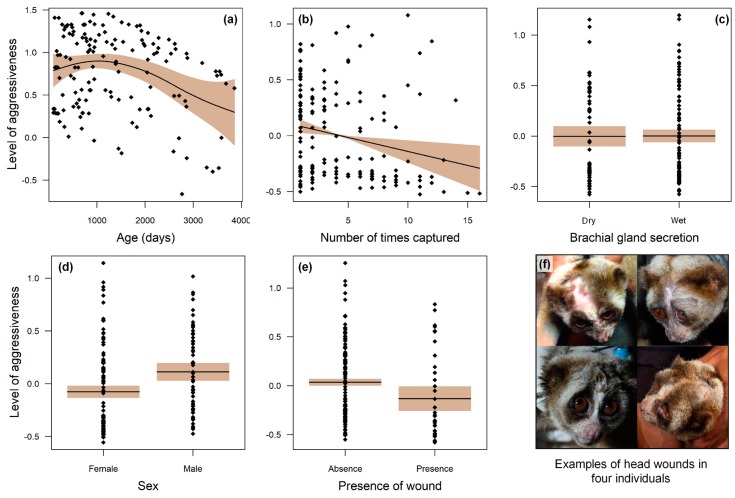
Relationship between level of aggressiveness with age (**a**), number of times captured (**b**), secretion of the brachial gland (**c**), sex (**d**), and presence of wounds (**e**) in a wild population of Javan slow lorises in Cipaganti, West Java. Y-axes are expressed as partial residuals related to the mean (µ = 0). The coloured area represents the 95% confidence level interval. Head wounds in four young slow lorises (**f**).

**Figure 5 toxins-11-00093-f005:**
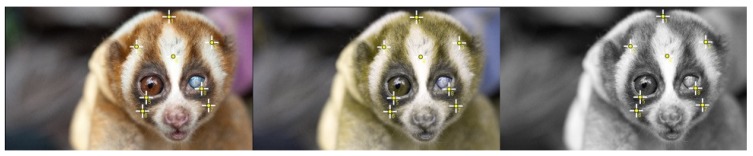
An adult Javan slow loris female in trichromatic, dichromatic and monochromatic views, showing 8 independent points on the facial mask from which we measured colour.

**Table 1 toxins-11-00093-t001:** Details on the drivers of Javan slow loris behaviour in relation to aggressiveness level in Java, Indonesia, according to the sex, age (days), presence of wound, degree of brachial gland secretion and the number of captures and recaptures the individual handled. The best-fitted model used the Inverse Gaussian family of distribution, yielding a generalised r^2^ = 031 and AIC = 217, with ΔAIC = 2.87 in relation to the second best-ranked model.

Model	Estimate	Std. Error	*t* Value	*p* Value
Aggressiveness level ~				
Sex	0.188	0.071	2.66	0.008 *
Age	−0.126	0.0005	2.19	0.030 *
Presence of wounds	−0.168	0.079	−2.12	0.035 *
Brachial gland secretion	−0.003	0.064	−0.06	0.951
Number of captures	−0.025	0.008	−2.87	0.004 *

* Denotes values less than 0.05.

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
