# Peer review of "Venom in Furs: Facial Masks as Aposematic Signals in a Venomous Mammal"

_toxins, 2019, doi:10.3390/toxins11020093_

Round 1
Reviewer 1 Report
GENERAL
This is an interesting but speculative paper about loris aposematism. I found much of the discussion highly speculative – that may be OK but I would like to have seen some evidence for many of the statements (see SPECIFIC below).
Many of the paragraphs cover a lot of ground very briefly and put a lot of different ideas next to each other (see SPECIFIC below). This makes for difficult reading.
Thus (i) writing style, and (ii) thoughtful assessment of dividing up their statements into those that are supported by definitive evidence, and those statements that are speculative, is required.
SPECIFIC
29 Brandley et al – stick to mammal references – Stankowich et al?
30 weaponry may be the defenses not the advert
32-35 masquerade and Mullerian mimicry are tangential – not needed here so omit this section
36-38 sentence unclear – omit
35-48 not clear what the paragraph says. Suggest reword or omit. Are you trying to say defenses and adverts are linked – if so I think you said that earlier.
59 see the giant panda eye mask in Caro et al 2017 too.
60-75 This is another rambling paragraph I am afraid. What are you trying to say here? That the same patterns have different functions? Why are you saying that?
76 what combination?
76-96 A lot to digest here – evidence that the loris mimics cobras – what was the nature of that evidence. Why do younger animals need stronger venom – does the venom kill other lorises? How do conspecifics assess venom toxicity – rather a lot of assumptions here that need careful spelling out.
103-111 spell out exactly why you make threse predictions – they are not crystal clear here.
115 what is the methodology behind these “schemes” needs a little explanation before they are introduced.
122 contrast index needs explanation before first mention
162 but colour does not distinguish them
176-177 rather a weak argument – omit
183 MAY disrupt it to some observers of some species
218 paragraph starts discussing intraspecific communication but then quickly turns to aposematism in other groups. This is a problem.
241-2 Very speculative. Furthermore contrast could simply be higher in cleaner fur of young animals
245 speculative again
250 now another idea from aposematism – kin recognition – this paragraph wanders a lot
260-263 again very speculative
271-274 I see no evidence for this assertion in the paper
Author Response
Please find our extensive revisions to the manuscript. We feel we could address every comment of the reviewers either through some relatively simple clarification but also through the suggested reanalyses of data. We hope that you will now find the manuscript suitable for publication and thank you and the reviewers for such a timely turn-around of the manuscript. We appreciate at this time of year it can be especially difficult to find the reviewers and we are really grateful for their extensive comments and attention.
We agree that this was a complex paper and we hope that some of the confusion the reviewers mention below is now clarified through rewriting and modifying key areas. We address all their concerns point-by-point.
This is an interesting but speculative paper about loris aposematism. I found much of the discussion highly speculative – that may be OK but I would like to have seen some evidence for many of the statements (see SPECIFIC below). – We feel that postulating on the function of the colour will always be speculative, but we try to provide more grounding with the most definitive results – see below.
Many of the paragraphs cover a lot of ground very briefly and put a lot of different ideas next to each other (see SPECIFIC below). This makes for difficult reading. We have tried to eliminate this – see below.
Thus (i) writing style, and (ii) thoughtful assessment of dividing up their statements into those that are supported by definitive evidence, and those statements that are speculative, is required. We have tried to make each paragraph more about definitive vs speculative; requiring further study – see below.
SPECIFIC
29 Brandley et al – stick to mammal references – Stankowich et al?
In this section we meant to discuss aposematic signals in general, since most venomous mammals do not have these signals; nonetheless we changed it and combined this and the next paragraph
30 weaponry may be the defenses not the advert; we have deleted weaponry – thank you!
32-35 masquerade and Mullerian mimicry are tangential – not needed here so omit this section - deleted
36-38 sentence unclear – omit – Combined with the next sentence
35-48 not clear what the paragraph says. Suggest reword or omit. Are you trying to say defenses and adverts are linked – if so I think you said that earlier. – We had intended to have one paragraph on signals and one on toxins – but agree there was overlap and have combined the two paragraphs.
59 see the giant panda eye mask in Caro et al 2017 too. – we have added this reference – it was useful many times
60-75 This is another rambling paragraph I am afraid. What are you trying to say here? That the same patterns have different functions? Why are you saying that? – We tried to say that the same patterns seen by animals with different visual systems might explain multiple functions of a signal. We have deleted some aspects and reworded and added a bit more about multi-modal signalling and the purpose of why we chose the different colour systems (also in response to R2)
76 what combination? – This referred to the last sentence in the previous paragraph, which listed a combination of functions. We have deleted that sentence and moved it here and hope it makes sense
76-96 A lot to digest here – evidence that the loris mimics cobras – what was the nature of that evidence. Why do younger animals need stronger venom – does the venom kill other lorises? How do conspecifics assess venom toxicity – rather a lot of assumptions here that need careful spelling out.
We have rearranged and amended this paragraph. As we have written several papers on this topic, we were worried to go into too much detail. We now deleted what we felt was unnecessary detail and replaced it with more pertinent details and tried to arrange the dparagraph, so it moves more neatly from predators to competitors. We also include more detail on the slow loris behaviour.
103-111 spell out exactly why you make threse predictions – they are not crystal clear here.
We have worked on the hypotheses and hope they are now clearer.
115 what is the methodology behind these “schemes” needs a little explanation before they are introduced. – I think the problem here is that the methods come after the results in this journal – they are explained in the methods – nonetheless we hope additions to paragraph above help to clarify
122 contrast index needs explanation before first mention – This is also in the methods (that appears after the results in this journal) – but we have included more on why we feel contrast is important and hope it clarifies
162 but colour does not distinguish them – changed the term colour to contrast – thank you for pointing this out
176-177 rather a weak argument – omit - omitted
183 MAY disrupt it to some observers of some species - added may
218 paragraph starts discussing intraspecific communication but then quickly turns to aposematism in other groups. This is a problem. – We have tried to make the slow loris section more firm – for us the dispersal and fighting IS a main aspect intraspecific competition so we restructured this paragraph deleting speculative aspects
241-2 Very speculative. Furthermore contrast could simply be higher in cleaner fur of young animals – It is definitely not that they are cleaner – it is just that they are almost black as opposed to the dark or reddish brown of adults and thus the contrast between white in the interocular striped compared to the eye patch is much more highly contrasting – this result was very significant in the model; we hope we have explained it better now – we also included a new statistical test on contrast and aggressiveness (R2) and hope this further supports this statement
245 speculative again – We agree that this is speculative and we have made it more clear that this is interesting and needs to be tested biochemically
250 now another idea from aposematism – kin recognition – this paragraph wanders a lot – we deleted this as we realised it does not really fit
260-263 again very speculative – see line 245 – tried to make this clearer as a future area of research
271-274 I see no evidence for this assertion in the paper – we have changed the term evolution to ecological function
Are the conclusions supported by the results? | ( ) | ( ) | (x) | ( ) |
Reviewer 2
This study investigates some biological roles of coloration in a toxic mammal. Authors captured 58 young loris in the wild and show that young slow loris had a more contrasted facial mask than adults. Other parameters of coloration did not differ between class ages. They also show that young loris were more aggressive than adults, and that more aggressive individuals had fewer wounds. However no link between color, aggressiveness and venomous gland secretion was found. Data and results seem interesting, and could potentially enrich the existing literature on the implications and biological roles of animal coloration. However, I feel that the work hypotheses are not clearly enough exposed and/or mixed up in the introduction and discussion, which makes the manuscript and conclusions difficult to follow. Statistical models could be improved or clarified to test some of the hypotheses. As a consequence, it is difficult to understand whether the presented results do support or not the authors’ hypotheses, which weakens the implications of the study. I also think that some conclusions should be down toned as some alternative explanations are possible and sample size is relatively low. To my opinion, a reorganization of the manuscript and a better clarification of the work hypotheses are needed and will help clarifying the implication of the study. Below are my detailed comments.
We thank the reviewer for these comments and we have clarified the hypotheses and also tried to reduce any speculative discussion. We should also note that we were working with a Critically Endangered mammal, and although our sample size is low compared to some other taxa, it is one of the largest studies done on a wild mammal so far spanning continuously over 8 years, and we hope that the editor can understand this constraint.
Main comments:
Aposematic signal and antipredator defense L21 and thereafter. In the key contribution part L21, authors claim “We present evidence that the contrasting facial mask of a venomous mammal, the Javan slow loris, is aposematic, used in intraspecific competition and as an anti-predator defense”.
We deleted the conclusion regarding anti-predator defense as this part of our work is more speculative. At the same time, in our continuous 8 years of study, we have not had a single incidence of predation despite a healthy potential predator community and we point this out in the methods.
However, there is no result in the study exploring predation risk. In addition, there was no link between color or contrast parameters and toxicity nor predation. I thus feel that authors cannot conclude that this trait is used as anti-predator defense.
We have toned down the anti-predator defense; indeed we only mentioned it as an ultimate cause (that would be difficult to test proximally), but we make that even more clear now.
Structure of the introduction and work hypotheses, from L45. Hypotheses about the roles of colors and age-specific differences are sometimes mixed in the introduction, which makes it difficult to understand which specific hypotheses will be tested by authors in the study. Authors should better clarify which hypotheses have already been validated by previous studies and which hypotheses have not yet been tested in the literature. I suggest better emphasizing the gaps in the literature about the role of facial masks in mammals, and to better present which specific hypotheses are tested by authors.
For instance, L45 the two hypotheses could be better explained. From what I understood, one hypothesis is that “The level of colouration or contrast may quantitatively indicate the level of toxicity“, which could be true both in adults and juveniles. The second hypothesis is that “colouration may be associated with younger animals that may either be more venomous or unable to meter their venom”, which predicts that juveniles would have more or less venom than adults, and consequently a different color. I suggest presenting hypotheses more explicitly to clarify the introduction. Maybe present hypotheses about the roles of colours and age-specific colours separately?
We have rewritten the hypotheses
From L60: this paragraph is interesting but several biological roles of colour in different contexts are presented altogether, making it difficult to understand what specific points are already validated in the literature or still unknown and which aspects the authors want to test in their study. For instance, authors present the potential role of the habitat background L71 (which is never tested and discussed in the rest study) and right after they present the potential role of color to direct bites towards less vulnerable body parts (which will be discussed in the study). This is confusing. I suggest better organizing the introduction in different paragraphs explaining more clearly and separetely the different roles of colors (even if they are not mutually exclusive) and the potential age differences and also explicitly emphasizing which specific hypotheses will be tested by authors in their study. To my opinion, this will help clarifying the manuscript.
-We have deleted the irrelevant sections and re ordered and rewritten all the introductory paragraphs.
Multimodal warning system and multiple signals: This aspect is rapidly exposed in the introduction L63 but could be developed more in depth, especially in discussion, because in your study you can test some potential combinations between color/ venom/behavior (aggressiveness).
We have added several new references here and tried to make it clearer why we test the colour systems and the contrast.
L 102. Different visual systems and anti-predatory role. The implications of your results regarding the different visual systems of predators (di or trichromatic) vs the loris (monochromatic) could be better emphasized in the discussion. At the end of the introduction L105, you explain that if the facial mask has a different antipredator function in young and adults, it should be distinguished in di or trichromatic schemes between age classes. It seems to me that this hypothesis was not validated by your results, which is interesting. However, in your discussion, you conclude that colors could have an antipredatory role.
This was more regarding contrast itself – we originally thought that different predators might see loris age classes differently but from a colour perspective they all look the same – it is only the contrast that is different and that a predator could still see.
We have clarified the hypotheses to make the contrast hypothesis stronger, and tone down the anti predator implications. We personally feel from observing the animals that the main role is intraspecific competition form a proximate perspective but we felt we could not rule out the predation aspect.
More generally, negative results are not discussed in the discussion, making it difficult to understand whether your results match your predictions or not. I am thus not convinced by some conclusions. Please discuss also your negative results and clarify how your work hypotheses are supported or not by your results. Also, some conclusions should be strongly down toned. See details below.
We try to make the questions clearer in the discussion.
Detailed comments:
L89. The paragraph about the model species is interesting. Are there already evidence of intraspecific variations in mask colors (within an age class) and/or in venom toxicity in this species?
We have modified this paragraph and tried to make it clearer. Unfortunately no information is available about venom toxicity in age classes other than the higher rate of death and injury among young and dispersing individuals and we cite articles regarding that. We do mention the mask resulting in the naming of two species (it is phenotypically very very different!)
L95. The fact that adults and young have been described as different species based on colour in the past is rather striking and important. I guess that this is from this observation that you built up some of your work hypotheses. If so, this aspect should be better emphasized and maybe presented earlier. You could start from this observation and then build up your hypotheses.
We have now changed the order and clarify this. – yes see above
L109. Predictions about mask roles and age specificity are a bit mixed up. Please clarify.
We hope these are clarified now and have rewritten them
L132. Figure 2 is quite convincing and interesting but the title could be clarified. The legend of Figure 2 (left) is confusing. I suppose that this represents NDMS of contrasts and not colours? Please correct or clarify the title and legend. What is “stress=0.010” on the figure? Please also add statistical values in the text or figure on contrast NDMS to justify the significant differences in contrast NDMS between juveniles and adults.
Thank you for pointing out this important mistake in the figures. We have now corrected caption. The important statistical value to report for NMDS is the stress. However, we also provide two complementary tests to obtain the p-value for the difference among ages, an ANOSIM, which tests the difference among the classes from the NMDS, and a post-hoc generalised additive model (GAM), to detect the relationship between the NMDS main Axes and the age as a continuous variable. For both we provide the respective p-value and additional statistical values required.
The stress value reflects how well the present ordination summarizes the distances among samples, i.e. how well fitted and reliable is the multi-dimensional model constructed. Lower stress values indicate greater conformity and therefore are desirable. High stress values indicate that there was no 2-dimensional arrangement of your points that reflect their similarities. A rule of thumb is that stress values up to 0.09 provide an excellent representation in reduced dimensions, stress values between 0.1 and 0.2 are considered good and stress values superior to 0.21 provide a poor representation. This detailed explanation is not usually included with an NMDS (a bit like explaining a p value) so we have not included it in the text for now but we include a reference.
L158. Discussion. Please discuss the fact that colours did not vary while contrasts did. Does this validate or invalidate some of your work hypotheses?
We have now included the hypotheses in each paragraph…and we point out this information about colour (also toning down the predation section)
L161. You find strong sex differences in behavior but this sex-specific aspect was not presented in the introduction. Did you have specific hypotheses on sex difference in colour and/or behavior/toxicity? How does this support some of the potential roles of signals? (sexual selection? Differences in competition?) we have added aspects of sex to the introduction – we did not expect it but we have an explanation for it and put some of that here (these are data that are also being prepared for a concurrent paper on territorial behaviour). We have added more in the discussion.
L166. Please explain how your results on colours in different visual systems validate or not your hypotheses on antipredatory role presented in the introduction. You present several predictions in the end of introduction but you don’t come back explicitely to these predictions in your discussion, which makes it confusing. From what I understood, the only result is that young and adults vary in the contrasts (and not colours) of their facial mask, but this result can be linked to several biological roles.
We have now addressed this hopefully by explicityly stating each hypothesis in the discussion
L170-190. Discussion about cryptism and disruptive coloration seems a bit out of scope. No clear hypothesis on cryptims was presented in the introduction and it seems to me that your data cannot test cryptism, because you cannot test the effects of colours on predation. In addition, you focus only on the facial mask in your study, so that other aspects of body colors were not tested. Please delete or considerably reduce this paragraph, and especially the parts on other body parts.
We have focussed these paragraphs back to the face mask and removed the back aspects other than to discuss the startle response
L198-235. I like this paragraph but I think that you should relate ot better to your results. It seems in particular that under the intraspecific competition hypothesis, you expected also a link between color features and venom, which was not supported by your results (no link between color or behavior and gland secretion Fig 3). I suggest better emphasizing which results support this hypothesis and which results don’t. Please also relate results to your predictions made in introduction.
Hopefully the new analysis and figure help to stabilise and reinforce this point.
L261. Here again it is difficult to understand to what extent your results support or not your hypotheses. You conclude “Thus, extreme contrast in juveniles could be a form of auto-mimicry, whereby smaller, more vulnerable juveniles can advertise their noxiousness by resembling larger more powerful adults”. I don’t understand how your results support this conclusion. How do juvenile resemble adults? Contrasts and behaviours differed between ages, and you did not find any link between behavior or toxicity and color in adults as well. I suggest clarifying how results support or not your hypotheses and down toning your conclusions.
We agree and have modified this section. We intended to show that young lorises want to appear larger and more powerful but in fact we realise this contradicted what we meant.
L263. Multimodal signaling and combinations between signaling traits (color/aggressiveness/toxicity) could be better discussed as some point (see also previous comment).
Added and connected with the introduction
L292. Explain better how you coded brachial secretion. It is an important point of the study but it is merely explained in methods. For instance you say that you just describe it as “dry” or “wet” but in Fig 3, there are 4 levels for this variable. Please clarify and explain how this can truly reflect toxicity, which is an important prerequisite of the study.
We thank the reviewer for the observation. We have replaced the figure using the two classes previously described in the methods (dry and wet).
L297. Please explain better how you measured aggressiveness. Several metrics can be used and are often combined to assess this behavioural trait (number of bite attempts, stereotyped behaviours, etc) but few explanation are given in your methods. In addition, L296, you explain that you assessed aggressiveness with 4 discrete levels (calm, restless but not aggressive, and aggressive), but in figure 3 it appears as a continuous variable. Please explain.
The level of aggressiveness was obtained and analysed at a discrete level (calm, feisty and aggressive – which we describe more in the text). However, generalised linear and additive models consider that all predictor variables are influencing the response variable simultaneously (in this case, level of aggressiveness) and, therefore, the graphical solution by each variable is a partial, in a residual scale. Therefore, although the response variable is discrete, its residuals are continuous.
Methods L 312. Color analyses and statistical models could be presented in separate paragraphs to clarify. Please explain better what RGB color and contrast reflect.
We have included several more statements about RGB and contrast in the introduction, with a bit extra in the methods and include separate paragraphs in the methods
L345 and Table 1. I don’t understand your choice of statistical model. It seems to me that you want to test whether contrast reflects toxicity and/or aggressiveness. Why not putting contrast as response variable? If I understand correctly you rather chose aggressiveness as response variable. Is then contrast included as explanatory variable in the model? In the methods L 345 you claim that contrast was included in the analysis but in the results it does not appear anymore. And what about age? Was it included? Maybe you included these variables but there were not kept in the best final model based on AIC? In this case, you could present the alternative models and the differences of AIC found. In all cases methods and results have to be improved and clarified.
Contrast is related to aggressiveness; however, many factors were also related to aggressiveness, such as sex, number of times the individuals were captured, body weight/age and activity of the brachial gland. Although we have data from 58 individuals, we could not photograph and measure the colour (tri-, di- and monochromatic) and contrast from all of them. The collection of these kind of data is labourious, and we had to limit our sample size for this factor. Therefore, to include the contrast in our model, we would reduce drastically the number of samples, which hampers the detection of significance for all the drivers of aggressiveness – especially because of fewer degrees of freedom. For instance, when we run a model to assess if the contrast is correlated to the levels of aggressiveness, we can clearly notice that individuals with greater contrast tend to be more aggressive, but the lower number of samples makes it not significant (E=-0.0026, SE=0.0016, t-value=1.42, p=0.18). Instead,we have added a new separate figure on contrast and level of aggressiveness. If we used the datasets together (as for those models generating the set of six panels), were reduced to 18 individuals, but by reducing the variables, we could use 31 individuals to run this analysis (Contrast Index Vs. Level of aggressiveness). We think the new results are really convincing that more contrasting individuals are indeed more aggressive.
To solve this issue, we used two collinear variables – body mass and age – as a proxy for contrast, allowing us to include all samples in the model. According to the generalised additive model performed as a post-hoc test for the NMDS, the contrast index is significantly correlated to the age of the individuals. In addition, age and body weight are also correlated. Therefore, in order to understand which variables are correlated to the aggressiveness, we included body weight, and after your suggestions, age as a proxy for contrast. We also included a discussion (L. 276 – 278) to explain these relationships.
L 345 and Table 1. You want to test whether color differ between ages. Why is age class not included as explanatory variable? Maybe your sample size is too low for so many explanatory variables, but maybe you could test separately test whether contrast or colour is linked to behavior and/or gland secretion separately in adults and juveniles? The rationale for that if that color could have different roles in juvenile and adults and be linked differently to other traits. In all cases methods and results have to be improved and clarified.
As you suggested, we now include age as an explanatory variable in the generalised additive models, and an ANOSIM to detect the relationship between the colours (tri-, di- and monochromatic scheme) or contrast and age (in L115-120 and figure 1 for colours and L 126 and figure 2 for contrast). We tested age as a continuous variable (in days) in the generalised additive models and as a categorical variable (age classes) in ANOSIM. Only contrast presented difference among age classes, decreasing with the age. Indeed, as we explained in the previous question, we cannot test all explanatory variables using the contrast due to the unavoidable reduction in the sample size, which reduces the degrees of freedom of the model and consequently the power and probability of the model in detecting significant relationships. See above the new figure that shows the relation between contrast and aggressive by reducing other variables in the model to increase the sample size.
Fig 3. Please illustrate also the difference in aggressiveness between ages, because this is an important result of your study.
We have not included age as an explanatory variable in the model because of the collinearity existent between age and body weight, and therefore, these two variables may be used interchangeably. However, we agree that age is an important variable to be shown, especially considering the relationship it has with contrast. To solve this issue, we have included the results for age in the table 1 and figure 1 and we added an observation in the methods to state that age and body weight are correlated.
Reviewer 2 Report
Venom in furs: Facial masks as aposematic signals in a venomous mammal. Toxins 422503
This study investigates some biological roles of coloration in a toxic mammal. Authors captured 58 young loris in the wild and show that young slow loris had a more contrasted facial mask than adults. Other parameters of coloration did not differ between class ages. They also show that young loris were more aggressive than adults, and that more aggressive individuals had fewer wounds. However no link between color, aggressiveness and venomous gland secretion was found. Data and results seem interesting, and could potentially enrich the existing literature on the implications and biological roles of animal coloration. However, I feel that the work hypotheses are not clearly enough exposed and/or mixed up in the introduction and discussion, which makes the manuscript and conclusions difficult to follow. Statistical models could be improved or clarified to test some of the hypotheses. As a consequence, it is difficult to understand whether the presented results do support or not the authors’ hypotheses, which weakens the implications of the study. I also think that some conclusions should be down toned as some alternative explanations are possible and sample size is relatively low. To my opinion, a reorganization of the manuscript and a better clarification of the work hypotheses are needed and will help clarifying the implication of the study. Below are my detailed comments.
Main comments:
Aposematic signal and antipredator defense L21 and thereafter. In the key contribution part L21, authors claim “We present evidence that the contrasting facial mask of a venomous mammal, the Javan slow loris, is aposematic, used in intraspecific competition and as an anti-predator defense”. However, there is no result in the study exploring predation risk. In addition, there was no link between color or contrast parameters and toxicity nor predation. I thus feel that authors cannot conclude that this trait is used as anti-predator defense.
Structure of the introduction and work hypotheses, from L45. Hypotheses about the roles of colors and age-specific differences are sometimes mixed in the introduction, which makes it difficult to understand which specific hypotheses will be tested by authors in the study. Authors should better clarify which hypotheses have already been validated by previous studies and which hypotheses have not yet been tested in the literature. I suggest better emphasizing the gaps in the literature about the role of facial masks in mammals, and to better present which specific hypotheses are tested by authors.
For instance, L45 the two hypotheses could be better explained. From what I understood, one hypothesis is that “The level of colouration or contrast may quantitatively indicate the level of toxicity“, which could be true both in adults and juveniles. The second hypothesis is that “colouration may be associated with younger animals that may either be more venomous or unable to meter their venom”, which predicts that juveniles would have more or less venom than adults, and consequently a different color. I suggest presenting hypotheses more explicitly to clarify the introduction. Maybe present hypotheses about the roles of colours and age-specific colours separately?
From L60: this paragraph is interesting but several biological roles of colour in different contexts are presented altogether, making it difficult to understand what specific points are already validated in the literature or still unknown and which aspects the authors want to test in their study. For instance, authors present the potential role of the habitat background L71 (which is never tested and discussed in the rest study) and right after they present the potential role of color to direct bites towards less vulnerable body parts (which will be discussed in the study). This is confusing. I suggest better organizing the introduction in different paragraphs explaining more clearly and separetely the different roles of colors (even if they are not mutually exclusive) and the potential age differences and also explicitly emphasizing which specific hypotheses will be tested by authors in their study. To my opinion, this will help clarifying the manuscript.
Multimodal warning system and multiple signals: This aspect is rapidly exposed in the introduction L63 but could be developed more in depth, especially in discussion, because in your study you can test some potential combinations between color/ venom/behavior (aggressiveness).
L 102. Different visual systems and antipredatory role. The implications of your results regarding the different visual systems of predators (di or trichromatic) vs the loris (monochromatic) could be better emphasized in the discussion. At the end of the introduction L105, you explain that if the facial mask has a different antipredator function in young and adults, it should be distinguished in di or trichromatic schemes between age classes. It seems to me that this hypothesis was not validated by your results, which is interesting. However, in your discussion, you conclude that colors could have an antipredatory role.
More generally, negative results are not discussed in the discussion, making it difficult to understand whether your results match your predictions or not. I am thus not convinced by some conclusions. Please discuss also your negative results and clarify how your work hypotheses are supported or not by your results. Also, some conclusions should be strongly down toned. See details below.
Detailed comments:
L89. The paragraph about the model species is interesting. Are there already evidence of intraspecific variations in mask colors (within an age class) and/or in venom toxicity in this species?
L95. The fact that adults and young have been described as different species based on colour in the past is rather striking and important. I guess that this is from this observation that you built up some of your work hypotheses. If so, this aspect should be better emphasized and maybe presented earlier. You could start from this observation and then build up your hypotheses.
L109. Predictions about mask roles and age specificity are a bit mixed up. Please clarify.
L132. Figure 2 is quite convincing and interesting but the title could be clarified. The legend of Figure 2 (left) is confusing. I suppose that this represents NDMS of contrasts and not colours? Please correct or clarify the title and legend. What is “stress=0.010” on the figure? Please also add statistical values in the text or figure on contrast NDMS to justify the significant differences in contrast NDMS between juveniles and adults.
L158. Discussion. Please discuss the fact that colours did not vary while contrasts did. Does this validate or invalidate some of your work hypotheses?
L161. You find strong sex differences in behavior but this sex-specific aspect was not presented in the introduction. Did you have specific hypotheses on sex difference in colour and/or behavior/toxicity? How does this support some of the potential roles of signals? (sexual selection? Differences in competition?)
L166. Please explain how your results on colours in different visual systems (mono-di-trichromatic) validate or not your hypotheses on antipredatory role presented in the introduction. You present several predictions in the end of introduction but you don’t come back explicitely to these predictions in your discussion, which makes it confusing. From what I understood, the only result is that young and adults vary in the contrasts (and not colours) of their facial mask, but this result can be linked to several biological roles.
L170-190. Discussion about cryptism and disruptive coloration seems a bit out of scope. No clear hypothesis on cryptims was presented in the introduction and it seems to me that your data cannot test cryptism, because you cannot test the effects of colours on predation. In addition, you focus only on the facial mask in your study, so that other aspects of body colors were not tested. Please delete or considerably reduce this paragraph, and especially the parts on other body parts.
L198-235. I like this paragraph but I think that you should relate ot better to your results. It seems in particular that under the intraspecific competition hypothesis, you expected also a link between color features and venom, which was not supported by your results (no link between color or behavior and gland secretion Fig 3). I suggest better emphasizing which results support this hypothesis and which results don’t. Please also relate results to your predictions made in introduction.
L261. Here again it is difficult to understand to what extent your results support or not your hypotheses. You conclude “Thus, extreme contrast in juveniles could be a form of auto-mimicry, whereby smaller, more vulnerable juveniles can advertise their noxiousness by resembling larger more powerful adults”. I don’t understand how your results support this conclusion. How do juvenile resemble adults? Contrasts and behaviours differed between ages, and you did not find any link between behavior or toxicity and color in adults as well. I suggest clarifying how results support or not your hypotheses and down toning your conclusions.
L263. Multimodal signaling and combinations between signaling traits (color/aggressiveness/toxicity) could be better discussed as some point (see also previous comment).
L292. Explain better how you coded brachial secretion. It is an important point of the study but it is merely explained in methods. For instance you say that you just describe it as “dry” or “wet” but in Fig 3, there are 4 levels for this variable. Please clarify and explain how this can truly reflect toxicity, which is an important prerequisite of the study.
L297. Please explain better how you measured aggressiveness. Several metrics can be used and are often combined to assess this behavioural trait (number of bite attempts, stereotyped behaviours, etc) but few explanation are given in your methods. In addition, L296, you explain that you assessed aggressiveness with 4 discrete levels (calm, restless but not aggressive, and aggressive), but in figure 3 it appears as a continuous variable. Please explain.
Methods L 312. Color analyses and statistical models could be presented in separate paragraphs to clarify. Please explain better what RGB color and contrast reflect.
L345 and Table 1. I don’t understand your choice of statistical model. It seems to me that you want to test whether contrast reflects toxicity and/or aggressiveness. Why not putting contrast as response variable? If I understand correctly you rather chose aggressiveness as response variable. Is then contrast included as explanatory variable in the model? In the methods L 345 you claim that contrast was included in the analysis but in the results it does not appear anymore. And what about age? Was it included? Maybe you included these variables but there were not kept in the best final model based on AIC? In this case, you could present the alternative models and the differences of AIC found. In all cases methods and results have to be improved and clarified.
L 345 and Table 1. You want to test whether color differ between ages. Why is age class not included as explanatory variable? Maybe your sample size is too low for so many explanatory variables, but maybe you could test separately test whether contrast or colour is linked to behavior and/or gland secretion separately in adults and juveniles? The rationale for that if that color could have different roles in juvenile and adults and be linked differently to other traits. In all cases methods and results have to be improved and clarified.
Fig 3. Please illustrate also the difference in aggressiveness between ages, because this is an important result of your study.

Author Response

(The authors gave the same response as above.)

Round 2
Reviewer 2 Report
I appreciate the authors’ efforts to address my previous comments, which clarified some parts, especially the introduction. The manuscript is thus improved but from my opinion, it still needs some rewriting and clarifications (especially in the discussion and conclusion parts). Please find below and attached my detailed comments and suggestions:
Detailed comments:
L16. The abstract is clearer, but your hypothesis on predation is not supported by your results and no link with toxicity was found. So please remove “reduce predation of juveniles” and “advertises toxicity”. To clarify, please change the sentence in the abstract L16 into “Change in colouration through development may play a role in intraspecific competition, and advertise aggressiveness to competitors and/or predators in juveniles.”
Introduction. L81-82. It is still difficult to understand what is already known from the literature and what you want to test in your study. Please clarify. For instance, you state L 81 that “Rather, the use of venom plays a more important proximate role in intraspecific competition”, but it was not demonstrated yet. It is rather one of your main work hypothesis that you want to test in your study. I suggest changing it into “In this study, we hypothesized that mask color and the use of venom plays a more important proximate role in intraspecific competition”
L85 “that and based»? Please correct.
L69 to 95. Although this new version of the introduction is much clearer, this part is a bit long and some hypotheses are still mixed “advertise toxicity/facilitate avoidance”. In addition, no clear mention of your hypothesis about a link with aggressiveness is presented. Most sentences are presenting the hypothesis that color signals toxicity.
L102. What do you mean by “noxiousness”? What is the difference with “toxicity”? I feel that a clear definition of what you mean by noxiousness and toxicity would help clarifying the hypotheses (see also below)
L100-102. The link between aggressiveness and toxicity is not clear. You write that “To examine if colour or age were related to toxicity, we recorded aggressiveness and gland secretions during capture as a proxy for noxiousness”. In addition you write L106. “Finally, we predicted that if the facial mask advertises higher degree of toxicity, more aggressive animals will show the most contrasting facial masks”. From what I understand, you say that aggressiveness reflects toxicity and this is a prerequisite of your work hypotheses. This is a strong prerequisite that is debatable and needs more justification. And it seems that your results do not support this prerequisite because more aggressive individuals did not have more gland secretions in your results (see Table 1). To my opinion, the only way to test the link between color and toxicity is to test the link between color and gland secretion, but this link was not significant, right? To me, aggressiveness and toxicity are two different traits, and aggressive individuals are not necessary more toxic. Please explain and justify better this point and explain what you mean by “toxicity”, “noxiousness” and “aggressiveness” before exposing your work hypotheses. Please then rewrite your hypotheses accordingly. Discussion and abstract might also need some rewriting to clarify this point.
L138 and 345. Please explain why you added body mass in your model. It seems to me that your hypotheses were about age and not mass. In addition, mass and age are likely to be strongly correlated, hence diminishing the reliability of your statistical model (collinearity).That might explain why “the level of aggressiveness decreases with an increase in the individual body size; AND juveniles to subadults were the most aggressive”. L138. Maybe remove body mass from explanatory variables if you added age instead? That might also increase the statistical power of your model. If you choose to keep both variables, please justify why.
L 145. Table 1. Are sex and mass linked?
L 156. Fig 3. I appreciate that the authors added Figure 3 that is nice and useful. However, please explain better which statistical model was used to test this link between contrast and aggressiveness.
L 167. Discussion: I appreciate that authors took into account my comments and linked better their results to their predictions at the beginning of each paragraph. This is clearer. However, some paragraphs still need rewriting, and I suggest reducing some parts.
L179-197. Again, I suggest strongly reducing this paragraph about predation and cryptism as explained in previous review, because it is too speculative and not supported by your results. The beginning is ok but the added paragraph in red is too long and out of scope.
L 199-210. I like this part, that is clear and supported by your results. However, make clearer which of your results support your hypotheses. For instance L 199 Add “In addition, individuals with a more contrasted mask were more aggressive”
L 227. “We should note that amount of brachial gland fluid secreted was not significant” This sentence is incomplete. Please rewrite and explain that this is not in accordance with your predictions (which is not a problem, but should be more clearly acknowledged).
L 245-259. This paragraph is confusing. Several explanations are mixed, making it difficult to follow. For instance, several hypotheses about “background mismatch L 251, “illusion of size” L 253 are presented, that are not linked with your results. I also don’t understand why you write that “young lorises advertise their venom more than adults”. Please clarify.
Conclusions L 261-277. The whole paragraph is very speculative, and contradictory with some of your results and discussion parts. Please summarize your main findings, and explain which hypotheses are supported or not more clearly. Please remove highly speculative parts. For instance remove the sentence L267 about the evolutionary explanation, and most of L 269-277. What do you mean by “changes in the definition of venom”?
L345 and Table 1. Statistics. Although I appreciate the author’s effort to clarify this part, I find it still unsufficiently justified. From your responses to my previous comment, I understand that you put aggressiveness as response variable, because too few data are available on color. And you then used body mass and age as a proxy of color as explanatory variables, again because too few data are available on color. It is confusing. First, you should explain this more clearly in the statistic method part, and make clear what your real sample size is to test this link (here 31 individuals if I understood well, and not 58 individuals). Second, you cannot conclude that color is linked to aggressiveness if you cannot directly include color (or contrast) in your model. Or maybe you did a separate analysis? Explanations L 344-352 are not clear concerning this point. Please explain better how you tested the link between color (contrast) and aggressiveness. And acknowledge the statistical limitations in the methods and/or in the discussion.
Author Response
We thank the second reviewer and editor for their comments and have addressed the detailed comments below and well as moved the Methods section from the end to the manuscript to just before the results for clarity.
L16. The abstract is clearer, but your hypothesis on predation is not supported by your results and no link with toxicity was found. So please remove “reduce predation of juveniles” and “advertises toxicity”. To clarify, please change the sentence in the abstract L16 into “Change in colouration through development may play a role in intraspecific competition, and advertise aggressiveness to competitors and/or predators in juveniles.”
We updated the abstract to reflect the reviewer’s suggestion, but kept our interest in toxicity for clarity.
Introduction. L81-82. It is still difficult to understand what is already known from the literature and what you want to test in your study. Please clarify. For instance, you state L 81 that “Rather, the use of venom plays a more important proximate role in intraspecific competition”, but it was not demonstrated yet. It is rather one of your main work hypothesis that you want to test in your study. I suggest changing it into “In this study, we hypothesized that mask color and the use of venom plays a more important proximate role in intraspecific competition”
Updated
L85 “that and based»? Please correct.
Corrected
L69 to 95. Although this new version of the introduction is much clearer, this part is a bit long and some hypotheses are still mixed “advertise toxicity/facilitate avoidance”. In addition, no clear mention of your hypothesis about a link with aggressiveness is presented. Most sentences are presenting the hypothesis that color signals toxicity.
We have revised this paragraph, by reducing it and being more explicit about out predictions related to toxicity and aggression.
L102. What do you mean by “noxiousness”? What is the difference with “toxicity”? I feel that a clear definition of what you mean by noxiousness and toxicity would help clarifying the hypotheses (see also below)
Here we use noxious and toxic interchangeably, but for clarity we have changed all instances of noxious to toxic.
L100-102. The link between aggressiveness and toxicity is not clear. You write that “To examine if colour or age were related to toxicity, we recorded aggressiveness and gland secretions during capture as a proxy for noxiousness”. In addition you write L106. “Finally, we predicted that if the facial mask advertises higher degree of toxicity, more aggressive animals will show the most contrasting facial masks”. From what I understand, you say that aggressiveness reflects toxicity and this is a prerequisite of your work hypotheses. This is a strong prerequisite that is debatable and needs more justification. And it seems that your results do not support this prerequisite because more aggressive individuals did not have more gland secretions in your results (see Table 1). To my opinion, the only way to test the link between color and toxicity is to test the link between color and gland secretion, but this link was not significant, right? To me, aggressiveness and toxicity are two different traits, and aggressive individuals are not necessary more toxic. Please explain and justify better this point and explain what you mean by “toxicity”, “noxiousness” and “aggressiveness” before exposing your work hypotheses. Please then rewrite your hypotheses accordingly. Discussion and abstract might also need some rewriting to clarify this point.
We have updated hypothesis to reflect us separating toxicity and aggression. In the actual analysis aggression and glad secretion are never used interchangeably and were treated as distinct variable, but reported as a proxy. We reviewed the abstract, results and discussion to ensure that the distinction was clear throughout.
L138 and 345. Please explain why you added body mass in your model. It seems to me that your hypotheses were about age and not mass. In addition, mass and age are likely to be strongly correlated, hence diminishing the reliability of your statistical model (collinearity).That might explain why “the level of aggressiveness decreases with an increase in the individual body size; AND juveniles to subadults were the most aggressive”. L138. Maybe remove body mass from explanatory variables if you added age instead? That might also increase the statistical power of your model. If you choose to keep both variables, please justify why.
We did have collinearity between body weight and age, therefore, we never include these variables in the same model, but we presented the results for them, using both variables interchangeably (L 356-357). Therefore, the model outcomes are really reliable and powerful, including the fact that younger individuals are indeed more aggressive than the older ones. We have removed body weight from the table for clarity.
L 145. Table 1. Are sex and mass linked?
There is no sexual dimorphism in this species and no link.
L 156. Fig 3. I appreciate that the authors added Figure 3 that is nice and useful. However, please explain better which statistical model was used to test this link between contrast and aggressiveness.
We provided with the text (L141 – 143) and figure 4 the relationship between the contrast and aggressiveness, which was significant (p=0.03). We also added a line making clear the test we applied for Colour/contrast Vs Aggressiveness (L 349-350). In addition, we have changed the methods to address all the remaining issues.
L 167. Discussion: I appreciate that authors took into account my comments and linked better their results to their predictions at the beginning of each paragraph. This is clearer. However, some paragraphs still need rewriting, and I suggest reducing some parts.
We have updated many parts of the discussion, improving clarity and reducing some parts.
L179-197. Again, I suggest strongly reducing this paragraph about predation and cryptism as explained in previous review, because it is too speculative and not supported by your results. The beginning is ok but the added paragraph in red is too long and out of scope.
We have reduced this paragraph and kept the discussion within the scope of our results as it relates to the slow loris.
L 199-210. I like this part, that is clear and supported by your results. However, make clearer which of your results support your hypotheses. For instance L 199 Add “In addition, individuals with a more contrasted mask were more aggressive”
We updated this paragraph and specified the results.
L 227. “We should note that amount of brachial gland fluid secreted was not significant” This sentence is incomplete. Please rewrite and explain that this is not in accordance with your predictions (which is not a problem, but should be more clearly acknowledged).
We updated this sentence to make it complete and address how our hypothesis fits into it.
L 245-259. This paragraph is confusing. Several explanations are mixed, making it difficult to follow. For instance, several hypotheses about “background mismatch L 251, “illusion of size” L 253 are presented, that are not linked with your results. I also don’t understand why you write that “young lorises advertise their venom more than adults”. Please clarify.
We have updated this paragraph to focus on how contrast in younger slow lorises may create the illusion of a larger size and why this pattern may persist in subadults and fade in adults.
Conclusions L 261-277. The whole paragraph is very speculative, and contradictory with some of your results and discussion parts. Please summarize your main findings, and explain which hypotheses are supported or not more clearly. Please remove highly speculative parts. For instance remove the sentence L267 about the evolutionary explanation, and most of L 269-277. What do you mean by “changes in the definition of venom”?
We have updated this paragraph to include a summary and to cut back on speculation.
L345 and Table 1. Statistics. Although I appreciate the author’s effort to clarify this part, I find it still unsufficiently justified. From your responses to my previous comment, I understand that you put aggressiveness as response variable, because too few data are available on color. And you then used body mass and age as a proxy of color as explanatory variables, again because too few data are available on color. It is confusing. First, you should explain this more clearly in the statistic method part, and make clear what your real sample size is to test this link (here 31 individuals if I understood well, and not 58 individuals). Second, you cannot conclude that color is linked to aggressiveness if you cannot directly include color (or contrast) in your model. Or maybe you did a separate analysis? Explanations L 344-352 are not clear concerning this point. Please explain better how you tested the link between color (contrast) and aggressiveness. And acknowledge the statistical limitations in the methods and/or in the discussion.
The only variables that would influence the level of contrast would be age and aggressiveness, and we did that separately (Figures 3 and 4). On the other hand, we additionally would like to understand the drivers of the aggressiveness rather than just the face level of contrast (such as the age, presence of wounds, sex or brachial gland secretion, etc). In this case, we did not include the contrast level as an explanatory variable because it was collinear to age and body weight, therefore, we must choose just one of the three to be used as a proxy for all. Since we had the option to choose between including level of contrast (reducing the sample to 31 individuals with all the behaviour measured) and age, we chose to keep age (as a proxy, providing a more robust model for all variables) and use this relationship to interpret the level of contrast as well. We changed the method to make clear the number of individuals we have for each variable. In addition, we make a statement to acknowledge our limitation in terms of number of individuals. In lines 185-188, we also explained the reason to choose the age as a proxy for the level of contrast.
Round 3
Reviewer 2 Report
The authors addressed most of my comments and I found this revised version of the manuscript much improved. Find below some last minor suggestions to improve langage and clarity of sentences.
L12-15. Please correct “We found that young slow lorises differed significantly from adults, being more contrasting and more aggressive, with aggressive animals showing fewer wounds. We suggest that aposematic facial masks… »
L22 : « is aposematic, and used »
L36 remove the coma after predators. I suggest adding an example of species.
L34-36. In these two sentences, are you talking about olfactory signals, or toxicity or both? They seem unrelated. I suggest adding L34 “strong olfactory signals or toxins» if this is what you meant.
L58. Cut the sentence in two parts for clarity. “…within its environment. Contrast is thus a universally accessible visual cue…”
L68 Please add also: “can also possibly serve”
L93. I suggest better emphasizing that this is one of your work hypothesis. ”unable to meter their venom, we hypothesized that a more contrasting facial mask may advertise their toxicity”
L95 and 96 and 99 and 100 (but not thereafter) “whether” instead of “if”.
L104: “signal” (present tense) instead of “signalled”. More generally, I suggest correcting and using the present or past tense more homogeneously throughout the manuscript. For instance L231 individuals “were” instead of “are”.
L105: “would” instead of could
L 255. Legend of Fig 5: Remove “The photographs show examples of head wounds 255 in the study population” which is now in Figure 4. The x axis has been truncated, making it difficult to read.
L257-268 Langage should be improved in this first paragraph of discussion.
L 257. “testing instead of “of”
L258. Cut the sentence in two parts: “…in a venomous mammal. We found strong evidence …”
L259. “Although colours themselves did not distinguish age classes under any of the three visual systems tested, contrast did.”
L261. I suggest being more explicit. “…younger animals had more contrasted facial masks than adults. This trait is potentially perceivable …”
L265. “However, the amount of brachial gland fluid was not related to any…”
L271. Add “was not supported, because colours themselves did not distinguish age classes under any of the three visual systems tested”
L281. Add “Further studies on predation risks are now needed to test this hypothesis”
L283. Cut the sentence. “was supported. In addition…”
L312 “showing the quality of venom »
L314 “this relationship was not statistically significant“
L321” More aggressive” individuals instead of “these distinct”?
L326 “that lack the attack power of adult”? please change this part
L339 “case study of”
L 339-359. The conclusion needs rewriting. It is too long, repetitive and some sentences are still unclear. Please be more synthetic and synthetize your main findings in a few sentences. Details on sample sizes are not useful here but and should be discussed in paragraphs above.
For instance L344-354 replace by “Here we found that young lorises had more contrasted facial masks and were more aggressive than adults. This supports the hypothesis that colouration patterns act as a form of communication among conspecifics. These contrasts could potentially also be perceived by predators, although further studies on predation risks are now needed. “
L354. The facial mask seems affective? I don’t understand what you mean by "affective". Please correct.
Author Response
We thank the reviewer for their further comments and have addressed them all updating the text for clarity and figure 5. We have highlight the amended sections of the paper in purple.
L12-15. Please correct “We found that young slow lorises differed significantly from adults, being more contrasting and more aggressive, with aggressive animals showing fewer wounds. We suggest that aposematic facial masks… »
updated
L22 : « is aposematic, and used »
updated
L36 remove the coma after predators. I suggest adding an example of species.
updated
L34-36. In these two sentences, are you talking about olfactory signals, or toxicity or both? They seem unrelated. I suggest adding L34 “strong olfactory signals or toxins» if this is what you meant.
updated
L58. Cut the sentence in two parts for clarity. “…within its environment. Contrast is thus a universally accessible visual cue…”
updated
L68 Please add also: “can also possibly serve”
updated
L93. I suggest better emphasizing that this is one of your work hypothesis. ”unable to meter their venom, we hypothesized that a more contrasting facial mask may advertise their toxicity”
updated
L95 and 96 and 99 and 100 (but not thereafter) “whether” instead of “if”.
updated
L104: “signal” (present tense) instead of “signalled”. More generally, I suggest correcting and using the present or past tense more homogeneously throughout the manuscript. For instance L231 individuals “were” instead of “are”.
updated
L105: “would” instead of could
updated
L 255. Legend of Fig 5: Remove “The photographs show examples of head wounds 255 in the study population” which is now in Figure 4. The x axis has been truncated, making it difficult to read.
updated
L257-268 Langage should be improved in this first paragraph of discussion.
updated
L 257. “testing instead of “of”
updated
L258. Cut the sentence in two parts: “…in a venomous mammal. We found strong evidence
…”
updated
L259. “Although colours themselves did not distinguish age classes under any of the three visual systems tested, contrast did.”
updated
L261. I suggest being more explicit. “…younger animals had more contrasted facial masks than adults. This trait is potentially perceivable …”
updated
L265. “However, the amount of brachial gland fluid was not related to any…”
updated
L271. Add “was not supported, because colours themselves did not distinguish age classes under any of the three visual systems tested”
updated
L281. Add “Further studies on predation risks are now needed to test this hypothesis”
updated
L283. Cut the sentence. “was supported. In addition…”
updated
L312 “showing the quality of venom »
Updated
L314 “this relationship was not statistically significant“
updated
L321” More aggressive” individuals instead of “these distinct”?
updated
L326 “that lack the attack power of adult”? please change this part
updated
L339 “case study of”
updated
L 339-359. The conclusion needs rewriting. It is too long, repetitive and some sentences are still unclear. Please be more synthetic and synthetize your main findings in a few sentences. Details on sample sizes are not useful here but and should be discussed in paragraphs above.
updated
For instance L344-354 replace by “Here we found that young lorises had more contrasted facial masks and were more aggressive than adults. This supports the hypothesis that colouration patterns act as a form of communication among conspecifics. These contrasts could potentially also be perceived by predators, although further studies on predation risks are now needed. “
updated
L354. The facial mask seems affective? I don’t understand what you mean by "affective". Please correct.
updated